# ADVERSARIAL LEARNING OF DECOMPOSED REPRE-SENTATIONS FOR TREATMENT EFFECT ESTIMATION

## ABSTRACT

Estimating the Individual-level Treatment Effect (ITE) from observational data is an important issue both theoretically and practically. Including all the pre-treatment covariates for prediction is unnecessary and may aggravate the issue of data unbalance. While the *confounders (C)* are necessary, there are some covariates that only affect the treatment (*instrumental variables, I*), and some only affect the outcome (*adjustment variables, A*). Theoretical analyses show that including extra information in $I$ may increase the variance lower bound, and hence should be discarded. To facilitate the decomposed representation learning for the ITE estimation, we provide a rigorous definition of $I, C, A$ in terms of the causal graph with an identifiability analyses. Under the guidance of a theoretical justification, we propose an effective ADR algorithm to learn the decomposed representations and simultaneously estimate the treatment effect by introducing adversarial modules to constrain the probabilistic relations. Our proposed algorithm can be applied to both categorical and numerical treatments and the disentanglement is assured by both theoretical analyses and empirical results. Experimental results on both synthetic and real data show that the ADR Algorithm is advantageous compared to the state-of-the-art methods. Theoretical analyses also provide a path to further explore the issue of decomposed representation learning for ITE estimation.

## 1 INTRODUCTION

The inference of individual treatment effect (ITE) is an important issue in causal inference and has a wide application in many decision-making scenarios, e.g., precision medicine (Jaskowski & Jaroszewicz, 2012; Alaa & Van Der Schaar, 2017; Alaa et al., 2017), individualized marketing (Sato et al., 2019; Wan et al., 2022), and personalized insurance products (Guelman et al., 2015). The *treatment* commonly refers to an intervention that can be actively determined (*not passively observed*) and we are concerned about the effect caused by the treatment for each individual.

There are two influential frameworks in causal inference: the potential outcomes framework proposed by Neyman and Rubin (Rubin, 1974; Splawa-Neyman et al., 1990), and the causal graph framework proposed by Judea Pearl (Pearl, 2009a). While different notations/operators are defined in each framework to formalize the "*treatment effect*" separately, they are equivalent by certain translations (Pearl, 2009b). In this paper, we adopt the potential outcomes framework to define the ITE as it does not require much additionally defined mathematical operators. Meanwhile, we adopt the causal graph framework to analyze the variables decomposition as it provides a more flexible and meaningful way to analyze different roles of the covariates.

In the potential outcomes framework, $Y_i(t)$ denotes the potential outcome that would be observed if unit $i$ received treatment $t$. The ITE refers to $Y_i(t) - Y_i(0)$. In practice, we estimate the ITE by the CATE(conditional average treatment effect), the best estimator of the ITE in terms of the mean squared error (Künzel et al., 2019). Compared to the standard supervised learning, the ITE estimation is more challenging since the counterfactual outcomes are unobserved and the treatment assignment might be confounded (Imbens & Rubin, 2015; Zhong et al., 2022). In the existence of confounders, the distribution of $Y(t)$ is commonly not equal to $Y|T = t$. To deal with the issue, the common practice is to introduce pre-treatment covariates such that $\{Y(t)|\boldsymbol{x}\} =_d \{Y|t, \boldsymbol{x}\}$ (*ignorability assumption*).

While Rubin (2008; 2009) suggest that including all the available pre-treatment covariates is a safe choice, the inclusion of unnecessary covariates may harm the accuracy of the ITE estimation.

Intuitively, introducing the covariates that only affect $T$ and not $Y$ will enlarge the discrepancy between $p(\boldsymbol{x}|T=1)$ and $p(\boldsymbol{x}|T=0)$ and dropping such covariates will not shake the ignorability assumption. Theoretically, Shalit et al. (2017) derives an upper bound of the mean squared error of the estimated ITE and shows that decreasing such distribution discrepancy is beneficial to lower this upper bound. In this paper, we also show that the variance lower bound of the conditional treatment effect (CATE) could be very large when the propensity score $p(T=1|\boldsymbol{x})$ is extreme.

Enlightened by the above idea, a set of representation learning-based deep learning methods have been proposed for ITE estimation. This line of methods can be divided into two classes: one is based on balanced representation learning (Shalit et al., 2017; Johansson et al., 2016; 2022), and the other is driven by decomposed representation learning (Hassanpour & Greiner, 2020; Zhang et al., 2021b; Wu et al., 2022). As for the first class, the model aims at learning a balanced representation $\Phi(\boldsymbol{x})$ and then uses the $\Phi(\boldsymbol{x})$ to predict the potential outcomes. This class of methods runs at risk of missing out on the information of necessary confounders and may undermine the ignorability assumption (Hassanpour & Greiner, 2019). To deal with this issue, Hassanpour & Greiner (2020); Zhang et al. (2021b) proposed to decompose the representations into three disentangled parts: the one that only affects $T$ (instrumental variables), the common cause of $T$ and $Y$ (confounders), and the one that only affects $Y$ (adjustment variables). As both Hassanpour & Greiner (2020) and Zhang et al. (2021b) could not guarantee the separation between different components, Wu et al. (2022) made further improvements regarding this point. However, the method in Wu et al. (2022) is designed for binary treatment and outcomes as its loss functions require calculating the IPM (Integral Probability Metric) of the learn representations between $T=1$ and $0$, as well as $Y=1$ and $0$. Besides, Wu et al. (2022) introduces a set of individual-level sample weights as parameters to learn, which may bring unbearable computational complexity for the large-scale data with a huge sample size.

To deal with the above issues, we propose the ADR (Adversarial learning of Decomposed Representations) algorithm, which is theoretically motivated by a preliminary analysis on the decomposition and has no requirements on the data types of $T$ and $Y$ in its applicability. We design two adversarial modules to constrain the probabilistic relations among the components, which is more flexible than the way of calculating IPM as adopted in Hassanpour & Greiner (2020) and Wu et al. (2022). The proposed ADR algorithm builds upon a rigorous and complete definition of the variables decomposition and is theoretically guaranteed by an identifiability analysis. Note that existing literature tends to describe the decomposition in an intuitive way, e.g., *"instrumental variables (I) are the ones that only affect the treatment"*. However, as long as $I \rightarrow T$ and $T \rightarrow Y$, the instrumental variables $I$ also affect $Y$ (in an indirect way). To avoid such vague interpretations, we provide a rigorous definition of the decomposition via the causal graph. On top of the graphical definition, we prove that such decomposition is identifiable and can be equivalently confined by a series of probabilistic constraints, then we show that such constraints can be learned by introducing adversarial modules.

To summarize, our main contributions are: (i) we propose the ADR algorithm for learning decomposed representations for the ITE estimation that is theoretically guaranteed and is empirically validated by experiments, and is directly applicable to both categorical and numerical treatment. (ii) we provide a rigorous definition of the variables decomposition via the causal graph and prove its identifiability, which has the potential of stimulating other practical algorithms; (iii) we show the benefit of variables decomposition by analyzing the non-parametric variance lower bound of the CATE estimand.

## 2 NOTATIONS AND PROBLEM SETUP

Let $T \in \mathcal{T}$ denote the treatment, $Y \in \mathcal{Y}$ denote the outcome, and $\boldsymbol{X} \in \mathcal{X}$ denote the pre-treatment covariates. Suppose that the observational data $\mathcal{D} = \{\boldsymbol{x}_i, y_i, t_i\}_{i=1}^n$, with $\{(\boldsymbol{X}_i, Y_i, T_i)\}$ identically distributed as $\mathbb{P}_{\boldsymbol{X}, Y, T}$. We adopt the Neyman-Rubin potential outcome framework (Rubin, 1974; Splawa-Neyman et al., 1990; Rubin, 2005) to define the treatment effect. For each treatment level $t \in \mathcal{T}$, let $Y_i(t)$ be the potential outcome that would have been observed when $T_i = t$. The individual treatment effect (ITE) for unit $i$ at treatment level $t$ is defined as

$$\tau_i^t = Y_i(t) - Y_i(0), \tag{1}$$

which is the difference between the potential outcome under $T=t$ and the control level $T=0$. In practice, we estimate the CATE (conditional average treatment effect) $\tau^t(\boldsymbol{x}) := \mathbb{E}[Y_i(t) - Y_i(0)|\boldsymbol{X} = \boldsymbol{x}]$, and use the estimation of $\tau^t(\boldsymbol{x}_i)$ to predict $\tau_i^t$. Künzel et al. (2019) shows that the CATE is the best estimator of the ITE in terms of the mean squared error. To connect the conceptually defined

*potential outcomes* with the observed variables, we make the following standard assumptions, which are commonly assumed in the literature (Imbens & Rubin, 2015).

**Assumption 1.** *(Consistency) The potential outcome $Y(t)$ of treatment $T = t$ equals to the observed outcome if the actual treatment received is $t$.*

**Assumption 2.** *(Ignorability) The potential outcome $Y(t)$ is independent with the assigned treatment $T$ conditional on the pre-treatment variables $\boldsymbol{X}$, i.e. $Y(t) \perp T | \boldsymbol{X}$.*

**Assumption 3.** *(Positivity) For any $t \in \mathcal{T}$, $p(t|\boldsymbol{x}) > 0$ for any $\boldsymbol{x} \in \mathcal{X}$ with $p(\boldsymbol{x}) > 0$.*

Under Assumptions 1 and 2, $\tau^t(\boldsymbol{x})$ can be expressed by observed outcomes as equation equation 2. Assumption 3 is to ensure there are available data to fit $\mathbb{E}[Y|\boldsymbol{x}, t]$ for all $t \in \mathcal{T}$ and $\boldsymbol{x} \in \mathcal{X}$.

$$\mathbb{E}[Y(t)|\boldsymbol{x}] = \mathbb{E}[Y(t)|\boldsymbol{x}, T = t] = \mathbb{E}[Y|\boldsymbol{x}, T = t] \Rightarrow \tau^t(\boldsymbol{x}) = \mathbb{E}[Y|\boldsymbol{x}, T = t] - \mathbb{E}[Y|\boldsymbol{x}, T = 0]. \quad (2)$$

Equation equation 2 suggests $\tau^t(\boldsymbol{x})$ is identifiable from observational data, where the 1st equal sign is due to the Assumption 2, and the 2nd one is from the Assumption 1.

As shown above, the *potential outcomes framework* provides a succinct way to formulate the *causal effect* by the potential outcomes. By contrast, the *causal graph* framework (Pearl, 2009a) requires more preliminary definitions before defining the causal effect (e.g., *do*-operator, *d*-separation, etc.), while it provides a more flexible way to analyze and select the covariates for adjustment. Luckily, these two powerful frameworks are equivalent by certain translations (Pearl, 2009b). The potential outcome $Y(t)$ may be equivalently defined by the $do-$operator in the causal graph and the "back-door" criterion provides a conceptually meaningful description for the Ignorability assumption. Specifically, when the covariates $\boldsymbol{X}$ block all the back-door paths from $T$ to $Y$, we have $Y(t) \perp T | \boldsymbol{X}$. The above discussion of the two frameworks also explains why we use the potential outcomes framework to define the ITE, and adopt the causal graph terms for the variables decomposition.

## 3 THEORETICAL ANALYSES

In this section, we firstly show the variance bound of the CATE to provide insights and motivations for the following variables decomposition. Then we formally define the *Instrumental variables, Confounders, and Adjustment variables* (abbreviated as $\boldsymbol{I}, \boldsymbol{C}, \boldsymbol{A}$) in the causal graph with an identifiability analysis, which allows us to analyze the probabilistic properties of each component and guides us to propose the decomposed representation learning Algorithm in section 4.

### 3.1 MOTIVATION: VARIANCE BOUND FOR THE CATE

Analyzing the variance lower bound for the targeted estimand can provide useful insights for proposing specific estimation methods. Before introducing the variance bound for the CATE, let us recall the classic Cramér-Rao inequality (Rao et al., 1992; Cramér, 1999), which provides a variance lower bound for the parametric model and measures the difficulty of estimating a certain parameter. Towards a similar purpose, Hahn (1998) derives the variance lower bound for the average treatment effect (ATE) and average treatment effect on the treated (ATT), a semi-parametric analog of the Cramér-Rao lower bound. Following the same method, we may derive the variance lower bound for the CATE. Theorem 3.1 shows the result for binary $T$, which can be readily generalized to multi-class categorical and numerical treatment by replacing the summation by integration (see Supplementary for details).

**Theorem 3.1.** *Let $\sigma_0^2(\boldsymbol{x}) = \mathrm{Var}(Y(0)|\boldsymbol{x})$, $\sigma_1^2(\boldsymbol{x}) = \mathrm{Var}(Y(1)|\boldsymbol{x})$ be the conditional variance, and $e(\boldsymbol{x}) = \mathbb{P}(T = 1|\boldsymbol{x})$ be the propensity score. Then for any $\sqrt{n}-$consistent estimation $\hat{\tau}(\boldsymbol{x})$, the lower bound for the asymptotic variance of $\hat{\tau}(\boldsymbol{x})$ is $\mathbb{V} := \mathbb{E}\left[\frac{\sigma_1^2(\boldsymbol{X})}{e(\boldsymbol{X})} + \frac{\sigma_0^2(\boldsymbol{X})}{1-e(\boldsymbol{X})}\right].$*

Theorem 3.1 suggests that the propensity score $e(\boldsymbol{X})$ is an important determinant of the variance bound as it enters in the denominator. When $e(\boldsymbol{X})$ is close to zero or one, the variance lower bound could be very large. Particularly, when $\sigma_1(\boldsymbol{X}) = \sigma_0(\boldsymbol{X}) \equiv \sigma^2$, we have $\mathbb{V} = \sigma^2/[e(\boldsymbol{X})(1 - e(\boldsymbol{X}))]$, thus $\mathbb{V}$ is minimized when $e(\boldsymbol{X}) = 1/2$. This result implies that, the more predictive information for $T$ the model includes, the more extreme the propensity score will become, and the larger $\mathbb{V}$ will be, which means an increase in difficulty in obtaining a precise estimation of the ITE. The results may also be appreciated from the perspective of data unbalance. When $p(t|\boldsymbol{x})$ approaches 0 or 1, the discrepancy between $p(\boldsymbol{x}|t = 0)$ and $p(\boldsymbol{x}|t = 1)$ will also be aggravated, which increases the

difficulty to predictive counterfactual outcomes. Therefore, instead of including as many pre-treatment covariates as we could (Rubin, 2008; 2009), it is more reasonable to decompose the covariates into different parts according to their roles and select the appropriate parts to estimate the ITE.

## 3.2 DEFINITIONS AND THEORETICAL ANALYSES OF $I, C, A$

Suppose the causal structure over $\boldsymbol{X} \cup \{T\} \cup \{Y\}$ is a directed acyclic graph (DAG), denote by $G$. Each node in $G$ represents a variable and each directed edge denotes a direct causal relation. Here we do not require the structure of $G$ as known and only use the causal graph to define the variables decomposition, and then derive the probabilistic relations from the graphical properties.

**Definition 3.1.** *Define Instrumental variables ($I$), Confounders ($C$), Adjustment variables ($A$) as*
$\boldsymbol{I} = \{X_i |$ *there exists an unblocked path from $X_i$ to $T$ and $X_i \notin PA(Y)$ and $X_i$ is not a collider*$\}$;
$\boldsymbol{C} = \{X_i |$ *there exists an unblocked path from $X_i$ to $T$ and $X_i \in PA(Y)\}$*;
$\boldsymbol{A} = \{X_i |$ *there exists an unblocked path from $X_i$ to $Y$, and no unblocked paths from $X_i$ to $T\}$,*
*where $PA(Y)$ denotes the set of parent nodes of $Y$.*

The definition is motivated by the intuitive idea that $\boldsymbol{I}, \boldsymbol{C}, \boldsymbol{A}$ are the variables set that cause only $T$, both $T$ and Y, and only $Y$, respectively (Hassanpour & Greiner, 2020). To appreciate this, we take the causal graph in 1a as an illustrating example. By Definition 3.1, $\boldsymbol{I} = \{X_1\}$ is the direct cause of $T$, $\boldsymbol{C} = \{X_2\}$ is the common cause of both $T$ and $Y$, and $\boldsymbol{A} = \{X_3\}$, which is in align with the intuitive motivation.

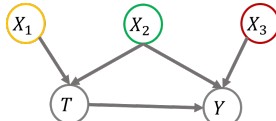
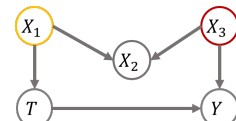

(a) *Illustrating Example 1*: $\boldsymbol{X} = (X_1, X_2, X_3)$ with $X_1, X_2, X_3$ being the instrumental variable, confounder, and adjustment variable, respectively.

(b) *Illustrating Example 2*: $\boldsymbol{X} = (X_1, X_2, X_3)$ with $\{X_1\}$ being the instrumental variable, and $X_3$ being the adjustment variable.

Figure 1: Illustrating Examples for Definition 3.1

In contrast to the intuitive description, Definition 3.1 provides a more specific and complete way to decompose the variables. In the example Figure 1b, it is easy to justify that $\boldsymbol{I} = \{X_1\}$, $\boldsymbol{C} = \emptyset$, $\boldsymbol{A} = \{X_3\}$, and $\boldsymbol{C} = \emptyset$ with Definition 3.1. The result $\boldsymbol{C} = \emptyset$ is also consistent with the fact that there are no unblocked backdoor paths from $T$ and $Y$. Formally, we may have the following general result.

**Proposition 3.1.** *Let $\boldsymbol{I}, \boldsymbol{C}, \boldsymbol{A}$ be the variables set defined in 3.1. Then (i) $\boldsymbol{C}$ blocks all the back-door paths from $T$ to $Y$; (ii) $P(Y|\boldsymbol{X}, do(t)) = P(Y|\boldsymbol{C}, \boldsymbol{A}, do(t))$*

Proposition 3.1 states including the *defined confounders $\boldsymbol{C}$* are sufficient such that $\mathbb{E}[Y(t)|\boldsymbol{C} \cup \boldsymbol{A}] = \mathbb{E}[Y|T = t, \boldsymbol{C} \cup \boldsymbol{A}]$ still holds by replacing $\boldsymbol{X}$ with $\boldsymbol{C} \cup \boldsymbol{A}$ in equation 2. The item (ii) means that $P(Y(t)|\boldsymbol{X}) = P(Y(t)|\boldsymbol{C}, \boldsymbol{A})$. That is, using $\boldsymbol{C} \cup \boldsymbol{A}$ to replace $\boldsymbol{X}$ would not lose the information for the inference of the individual treatment effect.

Moreover, the $\{\boldsymbol{I}, \boldsymbol{C}, \boldsymbol{A}\}$ in Definition 3.1 are identifiable without requiring further assumptions, which means the decomposition can be obtained from the joint distribution of $\{\boldsymbol{X}, T, Y\}$. Identifiability is crucial in statistics modeling because when the target is unidentifiable, it means we could not recover the true information even with infinite observations. The results are shown in Theorem 3.2.

**Theorem 3.2.** *The $\{\boldsymbol{I}, \boldsymbol{C}, \boldsymbol{A}\}$ are identifiable from the joint distribution $\mathbb{P}(\boldsymbol{X}, T, Y)$ as follows*
- *$X_i \in \boldsymbol{A} \Leftrightarrow \{X_i \perp T \text{ and } X_i \not\perp Y\}$*
- *$X_i \in \boldsymbol{I} \Leftrightarrow \{X_i \notin \boldsymbol{A}, X_i \not\perp T, \text{ and there exists a subset } \boldsymbol{X}' \subset \boldsymbol{X} \text{ s.t. } X_i \perp Y|\boldsymbol{X}' \cup \{T\}\}$*
- *$X_i \in \boldsymbol{C} \Leftrightarrow \{X_i \notin \boldsymbol{A} \text{ and } X_i \notin \boldsymbol{I} \text{ and } X_i \not\perp T \text{ and } X_i \not\perp Y\}$*

*Further, the confounders $\boldsymbol{C}$ may serve as the variables set $\boldsymbol{X}'$, i.e., $X_i \perp Y|\boldsymbol{C} \cup \{T\}$ for $X_i \in \boldsymbol{I}$.*

In brief, Proposition 3.2 states that
$$\{\boldsymbol{A} \perp T, \boldsymbol{A} \not\perp Y\}, \{\boldsymbol{I} \perp Y|\boldsymbol{C} \cup T, \boldsymbol{I} \not\perp T\}, \{\boldsymbol{C} \not\perp Y, \boldsymbol{I} \not\perp T\}, \tag{3}$$
and the three components have no overlaps. Theorem 3.2 shows the identifiability of $\{\boldsymbol{I}, \boldsymbol{C}, \boldsymbol{A}\}$ because the above equivalent conditions are in terms of the probabilistic relations instead of the graphical properties of $G$ (the structure of $G$ is commonly unidentifiable without further assumptions).

In practice, we learn the decomposed representations by the neural networks $\{I(\boldsymbol{X}), C(\boldsymbol{X}), A(\boldsymbol{X})\}$. The non-independent constraints in (3) are natural to implement by enforcing the predictive power of the learned representations. As for the (conditional) independent constraints $\boldsymbol{A} \perp T, \boldsymbol{I} \perp Y | \boldsymbol{C} \cup T$, Proposition 3.2 suggests that such properties can be constrained through an adversarial manner.

**Proposition 3.2.** *Denote $l(\cdot, \cdot)$ as the cross-entropy loss (categorical case) or $l_2$ loss (numerical case). Let $\hat{h}_{A \to T}(\cdot) := \arg\min_h l(h(A(X)), T)$ for given $A(\cdot)$, $\hat{h}_{C \cup T \to Y}(\cdot) := \arg\min_h l(h(\,C(X) \cup T), Y)$, $\hat{h}_{I \cup C \cup T \to Y}(\cdot) := \arg\min_h l(h(C(X) \cup I(X) \cup T), Y)$ for given $C(\cdot)$ and $I(\cdot)$. Then*

*(i) let $L_A := l(\hat{h}_{A \to T}(A(x)), T)$, then $L_A$ is maximized when $A(X) \perp T$;*

*(ii) let $L_{I,C} := l_d\big(\hat{h}_{C \cup T \to Y}(C(X) \cup T), \hat{h}_{I \cup C \cup T \to Y}(I(X) \cup C(X) \cup T)\big)$, where $l_d()$ denote the KL divergence (categorical $Y$) or $l_2$ loss (numerical $Y$), then $L_{I,C}$ is minimized when $I(X) \perp Y | \{T, C(X)\}$.*

Proposition 3.2 can be proved by firstly solving the $\hat{h}(\cdot)$'s and then substituting the expressions into $L_A$ and $L_{I,C}$ to prove the final result. Despite the heavy notations, the results of Proposition 3.2 can be interpreted from the intuitive perspective. Note that $A(X) \perp T \Leftrightarrow P(T|A(X)) = P(T)$, it means the optimal predictor of $T$ from $A(X)$ is uninformative, which implies $L_A$ should be maximized. Besides, $I(X) \perp Y | \{T, C(X) \Leftrightarrow P(Y|C(X) \cup I(X) \cup T) = P(Y|C(X) \cup T)$, which means $I(X) \cup T$ and $C(X) \cup I(X) \cup T$ have the same information for predicting $Y$. Thus the two optimal predictors should be the same and the distance $L_{I,C}$ is minimized. Please refer to the Supplementary for the detailed proof of Prop. 3.1, Prop. 3.2, and Prop. 3.2.

Building upon Proposition 3.2 and 3.2, we propose the following ADR algorithm to learn the decomposed representations through an adversarial manner as was adopted in GAN (Goodfellow et al., 2014), where $\{A(\cdot), I(\cdot), C(\cdot)\}$ and the predictors $\{h_{A \to T}(\cdot), h_{C \cup T \to Y}(\cdot), h_{I \cup C \cup T \to Y}(\cdot)\}$ play similar roles as the *generator* and *discriminator* in GAN, respectively.

## 4 ADR ALGORITHM

In this section, we introduce the ADR (*A*dversarial learning of *D*ecomposed *R*epresentations) algorithm, which learns the $\{I(\boldsymbol{X}), C(\boldsymbol{X}), A(\boldsymbol{X})\}$ and simultaneously predict the potential outcomes for the ITE estimation. The ADR algorithm is applicable for both categorical and numerical treatment.

### 4.1 OVERVIEW

Figure 2 demonstrates the modules required for the ADR algorithm. The module $f_{C \cup A \cup T \to Y}(\cdot)$ is used to predict the potential outcome $\hat{Y}(t)$, the modules $\{I(\cdot), C(\cdot), A(\cdot)\}$ are three parallel networks to learn the decomposed representations, and other modules $\{h_*(\cdot)\}$ and $f_*(\cdot)$ are designed to constrain the probabilistic relations stated in Theorem 3.2 (here $*$ denotes a placeholder, e.g., $h_*$ may denote $h_{A \to T}$ or $h_{C \cup T \to Y}$). The $\{\mathcal{L}_I, \mathcal{L}_C, \mathcal{L}_A\}$ and $\{\mathcal{L}_A^h, \mathcal{L}_I^h\}$ are components of the loss functions.

Overall speaking, the training process involves two iterative steps: (i) fix the representation networks and update the ancillary predictors $\{h_*(\cdot)\}$ by minimizing $\mathcal{L}_A^h + \mathcal{L}_I^h$. (ii) fix $\{h_*(\cdot)\}$ and update the representations $\{I(\cdot), C(\cdot), A(\cdot)\}$ and predictors $\{f_*(\cdot)\}$ by minimizing $\mathcal{L}_I + \mathcal{L}_C + \mathcal{L}_A$ plus the regularization loss terms, where each component corresponds to the constraints of each representation.

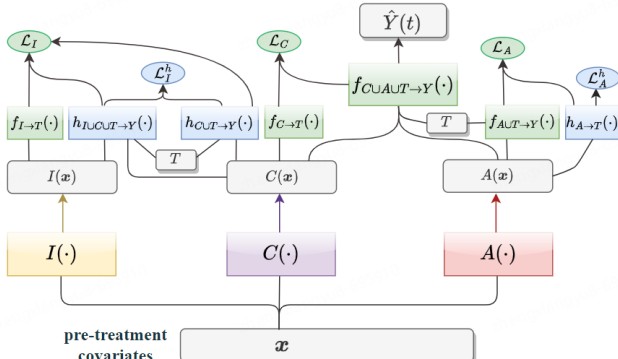

Figure 2: The model architecture and loss components of the ADR Algorithm (rectangles denote the neural networks; gray and rounded rectangles denote the inputs/outputs; ellipses denote the loss components)

### 4.2 LOSS FUNCTIONS FOR DECOMPOSED REPRESENTATIONS

In the following, we introduce the loss functions for each decomposed representation in details.

***Adjustment Variable***: **(i)** To realize $A(\boldsymbol{X}) \perp T$, we introduce an ancillary predictor $h_{A \to T}(\cdot)$ that predicts $T$ from $A(\boldsymbol{X})$ as suggested by Prop.3.2. We firstly fix $A(\cdot)$ and update $h_{A \to T}(\cdot)$ by minimizing $\mathcal{L}_A$ in (4), and then fix $h_{A \to T}(\cdot)$ to update $A(\cdot)$ by maximizing $\mathcal{L}_A$. **(ii)** As for $A(\boldsymbol{X}) \not\perp Y$, we introduce the predictor $f_{A \cup T \to Y}(\cdot)$ and minimize its loss in updating $A(\cdot)$. In summary, define

$$
\begin{aligned}
\mathcal{L}_A^h &:= \sum\nolimits_i l\left(h_{A \to T}(A(\boldsymbol{x}_i)), t_i\right), \\
\mathcal{L}_A &:= \sum\nolimits_i l\left(f_{A \cup T \to Y}(A(\boldsymbol{x}_i)), t_i\right) - l\left(h_{A \to T}(A(\boldsymbol{x}_i)), t_i\right) = \sum\nolimits_i l\left(f_{A \cup T \to Y}(A(\boldsymbol{x}_i)), t_i\right) - \mathcal{L}_A^h.
\end{aligned}
\tag{4}
$$

***Instrumental Variable***: **(i)** To realize $I(\boldsymbol{X}) \perp Y | C(\boldsymbol{X}) \cup T$, we introduce two ancillary predictors $h_{I \cup C \cup T \to Y}(\cdot)$ and $h_{C \cup T \to Y}(\cdot)$, which are firstly updated for given $I(\cdot)$ and $C(\cdot)$. Then we update the representations to minimize the discrepancy $l_d(\cdot, \cdot)$ between the two predictors, where the $l_d(\cdot, \cdot)$ refers to the KL-divergence for categorical $Y$ and the $l_2$ loss for numerical $Y$. **(ii)** For $I(\boldsymbol{X}) \not\perp T$, we include the predictor $f_{I \to T}(\cdot)$ and minimize its loss in updating $I(\cdot)$. In summary, define

$$
\begin{aligned}
\mathcal{L}_I^h &:= \sum\nolimits_i l\left(h_{I \cup C \cup T \to Y}\left(I(\boldsymbol{x}_i), C(\boldsymbol{x}_i), t_i\right), y_i\right) + \sum\nolimits_i l\left(h_{C \cup T \to Y}\left(C(\boldsymbol{x}_i), t_i\right), y_i\right) \\
\mathcal{L}_I &:= \sum\nolimits_i l\left(f_{I \to T}(I(\boldsymbol{x}_i)), t_i\right) + l_d\left(h_{I \cup C \cup T \to Y}\left(I(\boldsymbol{x}_i), C(\boldsymbol{x}_i), t_i\right), h_{C \cup T \to Y}\left(C(\boldsymbol{x}_i), t_i\right)\right).
\end{aligned}
\tag{5}
$$

***Confounders***: **(i)** To realize $C(\boldsymbol{X}) \not\perp Y$ and simultaneously predict the potential outcome $Y(t)$, we introduce the prediction module $f_{C \cup A \cup T}(\cdot)$ to predict $Y(t)$ from $\{C(\boldsymbol{X}), A(\boldsymbol{X}), T\}$. In implementation, we model $f_{C \cup A \cup T}(\cdot)$ in a two-model manner for binary $T$. **(ii)** To constrain $C(\boldsymbol{X}) \not\perp T$, we add a module $f_{C \to T}(\cdot)$ and minimize its predictive loss in updating $C(\cdot)$. In summary, define $\mathcal{L}_C$ as

$$
\mathcal{L}_C := \sum\nolimits_i l\left(f_{C \cup A \cup T \to Y}(C(\boldsymbol{x}_i), A(\boldsymbol{x}_i), t_i), y_i\right) + \sum\nolimits_i l\left(f_{C \to T}(C(\boldsymbol{x}_i)), t_i\right).
\tag{6}
$$

In addition to the loss functions above, we also require the following regularization components $\mathcal{L}_O$ and $\mathcal{L}_R$ to constrain the orthogonality of representations and to penalize the model complexity.

***Regularization Part***: **(i)** Following Wu et al. (2022); Kuang et al. (2017; 2020), we constrain $I(\boldsymbol{X}), C(\boldsymbol{X}), A(\boldsymbol{X})$ to be decomposed parts by imposing the following orthogonality constraint:

$$
\mathcal{L}_O = \left[\bar{W}_I^T \cdot \bar{W}_C^T + \bar{W}_C^T \cdot \bar{W}_A^T + \bar{W}_A^T \cdot \bar{W}_I^T\right] + \sum\nolimits_{\bar{W} \in \{\bar{W}_I, \bar{W}_C, \bar{W}_A\}} \left[\left(\sum\nolimits_{k=1}^m \bar{W}[k] - 1\right)^2\right]
\tag{7}
$$

where $\bar{W}_I$ is the vector obtained by averaging each row of $W_I$, and $W_I := W_{I1} \times \cdots \times W_{Ij} \times \cdots \times W_{Im}$ with $W_{Ij}$ as the weight matrix in the $j$-th layer of the representation network of $I(\cdot)$, and $\bar{W}_I[k]$ denote the $k$-th element of $\bar{W}_I$. Also, $\bar{W}_C, \bar{W}_C[k], \bar{W}_A, \bar{W}_A[k]$ are defined in the same way. **(ii)** To prevent overfitting, we add $l_2$ regularization on the weight parameters of prediction modules:

$$
\mathcal{L}_R = l_2(\mathcal{W}(f_{C \cup A \cup T \to Y}, f_{A \cup T \to Y}, f_{I \to T}, f_{C \to T})).
\tag{8}
$$

### 4.3 ADVERSARIAL LEARNING OF DECOMPOSED REPRESENTATIONS

In summary, let $\mathcal{L}^h := \mathcal{L}_A^h + \mathcal{L}_I^h$ and $\mathcal{L} := \mathcal{L}_C + \alpha \cdot \mathcal{L}_A + \beta \cdot \mathcal{L}_I + \mu \cdot \mathcal{L}_O + \lambda \cdot \mathcal{L}_R$, where $\{\alpha, \beta, \mu, \lambda\}$ are hyper-parameters to scale different loss components. We learn the decomposed representations via an adversarial process to update the parameters iteratively, which are summarized in Algorithm 1. Please refer to the supplementary material for the source code and the selection of hyper-parameters.

---

**Algorithm 1** Adversarial learning of Decomposed Representations

---

**Input:** observational data $\{\boldsymbol{x}_i, t_i, y_i\}_{i=1}^n$
**Output:** $\hat{y}_i(t)$ and $\hat{\tau}_i^t$ for $t \in \mathcal{T}$.
1: **for** the number of training iteratios **do**
2:     **for** $k = 1, \cdots, K$ **do**
3:         calculate loss $\mathcal{L}^h = \mathcal{L}_A^h + \mathcal{L}_I^h$;
4:         update the parameters of $\{h_{A \to T}(\cdot), h_{I \cup C \cup T \to Y}(\cdot), h_{C \cup T \to Y}(\cdot)\}$ by descending the gradient of $\mathcal{L}^h$
5:     **end for**
6:     calculate the main loss $\mathcal{L} = \mathcal{L}_C + \alpha \cdot \mathcal{L}_A + \beta \cdot \mathcal{L}_I + \mu \cdot \mathcal{L}_O + \lambda \cdot \mathcal{L}_R$.
7:     update $\{I(\cdot), C(\cdot), A(\cdot), f_{C \cup A \cup T \to Y}(\cdot), f_{A \cup T \to Y}(\cdot), f_{I \to T}(\cdot), f_{C \to T}(\cdot)\}$ by the gradient of $\mathcal{L}$
8: **end for**
9: calculate $\hat{y}_i(t) = f_{C \cup A \cup T \to Y}(C(\boldsymbol{x}_i), A(\boldsymbol{x}_i), t_i)$.
10: calculate the ITE estimation $\hat{\tau}_i^t = \hat{y}_i(t) - \hat{y}_i(0)$.

---

## 5 EXPERIMENTS

In this section, we report the performance of the proposed ADR algorithm on both aspects of the decomposed representation learning and the ITE estimation by synthetic and real datasets. The results show that the ADR algorithm is able to learn the correct decomposition of variables on the synthetic dataset under both binary and continuous data settings, with an ablation study to show the contribution of the theory-based adversarial loss modules empirically. As for the performance of ITE estimation, ADR also shows an advantageous performance with higher qini score (Zhang et al., 2021a) for binary outcomes and lower $\epsilon_{\text{PEHE}}$ (expected Precision in Estimation of Heterogeneous Effects (Shalit et al., 2017; Olaya et al., 2020)) for continuous outcomes. All experiments were conducted with one NVIDIA Tesla P40 GPU.

### 5.1 COMPARED METHODS AND EVALUATION METRICS

We compare the proposed **ADR** algorithm with the following representation learning-based methods:

- **CFR-MMD** and **CFR-WASS** (Shalit et al., 2017; Johansson et al., 2016): Counterfactual Regression with MMD and Wassertein metrics to learn the balanced representation.
- **CFR-ISW** (Yao et al., 2018): Counterfactual Regression with Importance Sampling weights.
- **DR-CFR** (Hassanpour & Greiner, 2020): Disentangled Representations for CounterFactual Regression, which includes the distribution metrics $Disc(\mathbb{P}(A|T=1), \mathbb{P}(A|T=0))$ and the predictive loss of $\{I(\boldsymbol{x}) \cup C(\boldsymbol{x}) \to Y\}$ in the loss function to drive the representations decomposition.
- **TEDVAE** (Zhang et al., 2021b): A VAE-based method that includes the ELBO and the predictive loss of $\{I(\boldsymbol{x}) \cup C(\boldsymbol{x}) \cup T \to Y\}$ and $\{A(\boldsymbol{x}) \cup C(\boldsymbol{x}) \to T\}$ to learn the representations.
- **DER-CFR** (Wu et al., 2022): Decomposed Representations for Counterfactual Regression, which includes $Disc(\mathbb{P}(A|t=1), \mathbb{P}(A|t=0))$, $\sum_t Disc(\tilde{\mathbb{P}}(C|Y=1, T=t), \tilde{\mathbb{P}}(A|Y=0, T=t))$ in the loss, with $\tilde{\mathbb{P}}$ as the data distribution re-weighted with sample weights $\{\omega\}$ as trainable parameters.

In summary, CFR-MMD, CFR-WASS, and CFR-ISW learn balanced representations to estimate the potential outcomes, while DR-CFR, TEDVAE, DER-CFR, and our proposed ADR learn decomposed representations and then use the confounders and adjustment variables to estimate the potential outcomes. Note that both DR-CFR and DeR-CFR require binary $T$ and even binary $Y$ (DeR-CFR) in calculating the distribution metrics, we transform $T$ or $Y$ into binary by setting the median as the thresholds, which was also the way adopted in the source code of Wu et al. (2022). To facilitate a fair comparison, all the representation-learning based methods share the same value of representation dimension and the same prediction head in our experiments. For the detailed parameters setting, please refer to the `configs/params_all.json` in Supplementary for details.

**Evaluation Metrics:** For the case with continuous $Y$ and ground truth ITE, we use the expected Precision in Estimation of Heterogeneous Effect $\epsilon_{\text{PEHE}} = \{\frac{1}{n}\sum_{i=1}^n [\hat{\tau}(\boldsymbol{x}_i) - \tau_i]^2\}^{1/2}$ (Shalit et al., 2017). For the case with binary $Y$ and without ground truth ITE, we use Qini score (the normalized area under the qini curve, Zhang et al. (2021a)) to evaluate the rank performance of the estimated ITE on the randomized controlled trial (RCT) data.

### 5.2 EXPERIMENTS ON SYNTHETIC DATASETS

To investigate the performance of decomposed representation learning, we conducted experiments on the synthetic data where the ground truth of $\{\boldsymbol{I}, \boldsymbol{C}, \boldsymbol{A}\}$ is known. To maintain consistent results and ensure a fair comparison, we directly adopt the synthetic dataset provided by Wu et al. (2022). Although Wu et al. (2022) only shows the results for binary setting in the paper, their data generation source codes provide both settings for binary and continuous $\{T, Y\}$.

The data generating process is as follows. The instrument variable $\boldsymbol{X}_I = (X_1, \cdots, X_{16})$, the confounders $\boldsymbol{X}_C = (X_{17}, \cdots, X_{32})$, and the adjustment variables $\boldsymbol{X}_A = (X_{33}, \cdots, X_{48})$. In addition to the above components, the covariates $\boldsymbol{X}$ also include $m_D = 2$ extra dimensions of noise variables. The covariates are independently generated from standard Normal distribution $N(0, 1)$. The sample size $n = 3000$. Let $\boldsymbol{X}_{IC} = (X_I, X_C)$ for generating $T$ and $\boldsymbol{X}_{CA} = (X_C, X_A)$ for generating $Y$.

***Binary Setting***: For the setting with binary treatment and binary outcomes,

- generate $t$: $T \sim B(1, p(\boldsymbol{x}_{IC}))$, where $p(\boldsymbol{x}_{IC}) = [1 + exp(-(\theta_t^T \boldsymbol{x}_{IC} + \varepsilon))]^{-1}$ with $\varepsilon \sim N(0, 0.1^2)$.

- generate $y$: Firstly, generate $\mu_0 = \theta_{y0}^T \boldsymbol{x}_{CA}$ and $\mu_1 = \theta_{y1}^T (\boldsymbol{x}_{CA} \cdot \boldsymbol{x}_{CA})$. Then, generate binary outcomes $y(1) = \text{sign}(\max(0, \mu_0 - \tilde{\mu}_0))$ and $y(0) = \text{sign}(\max(0, \mu_1 - \tilde{\mu}_1))$, where $\tilde{\mu}_0$ and $\tilde{\mu}_1$ denote the median numbers. Then, generate $y$ as $y(1)t + y(0)(1 - t)$.

***Continuous Setting***: For the setting with continuous treatment and continuous outcomes,

- generate the treatment $t = p(\boldsymbol{x}_{IC})$, where $p(\boldsymbol{x}_{IC})$ is the same as the binary setting.
- generate $y = y(t) = \mu_0 + \mu_1 \times t + \varepsilon$ with $\varepsilon \sim N(0, 0.1^2)$, where $\mu_0$ and $\mu_1$ are defined as above.

We compared ADR with DR-CFR and DeR-CFR on the performance of decomposed representation learning by $\{\bar{W}_I, \bar{W}_C, \bar{W}_A\}$ in equation (7), the average contribution of each element of $\boldsymbol{X}$ for $I(\cdot)$, $C(\cdot)$, $A(\cdot)$, respectively. According to the data generating process, the non-zero elements of $\bar{W}_I$ should be mainly on the first 16 variables because the ground truth is $\boldsymbol{X}_I = (X_1, \cdots, X_{16})$. Similarly, $\bar{W}_C$ and $\bar{W}_A$ are expected to concentrate on the middle 16 and the last 16 variables. Figure 3 shows the values of $\{\bar{W}_I, \bar{W}_C, \bar{W}_A\}$ by histograms for each algorithm in the binary case. Both ADR and DeR-CFR could approximately distinguish different partitions and DR-CFR fails to identify the decomposed representations, which is in align with the results reported in Wu et al. (2022).

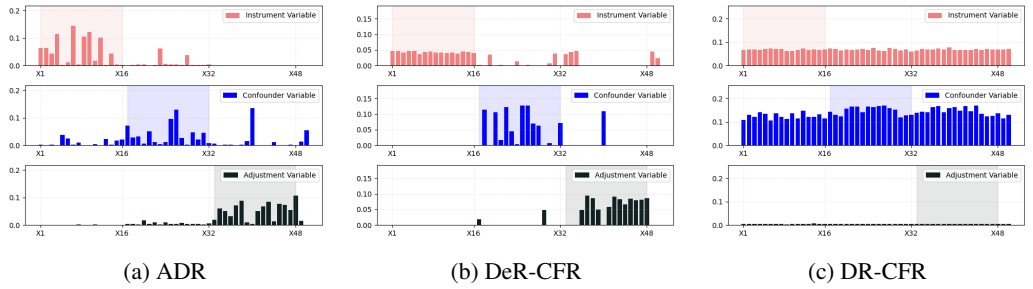

(a) ADR        (b) DeR-CFR        (c) DR-CFR

Figure 3: The $\{\bar{W}_I, \bar{W}_C, \bar{W}_A\}$ for the representation networks $\{I(\cdot), C(\cdot), A(\cdot)\}$ for the binary case

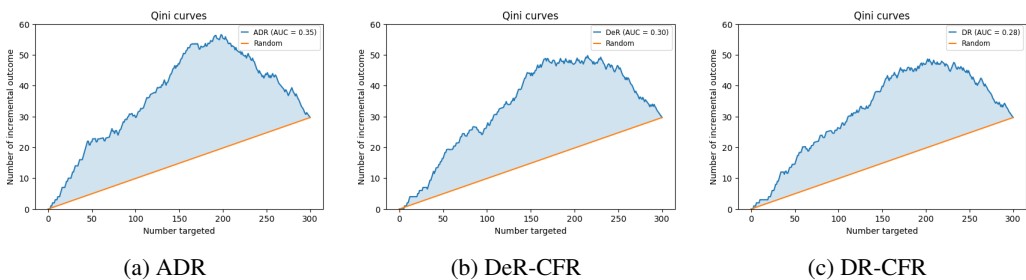

(a) ADR        (b) DeR-CFR        (c) DR-CFR

Figure 4: The qini curve based on the ITE estimation on the RCT data (sample size 300).

Figure 4 shows the qini curve (by `sklift package`) for the three methods. ADR also attains a higher qini score (0.35) than the DeR-CFR(0.30) and DR-CFR(0.23). Figure 5 shows the results for the continuous case, where ADR is still able to distinguish different components of the covariates approximately, but both DeR-CFR and DR-CFR fail to learn the correct decomposition.

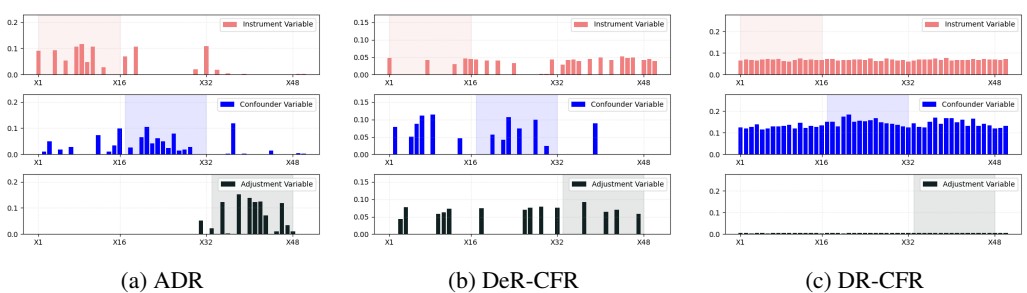

(a) ADR        (b) DeR-CFR        (c) DR-CFR

Figure 5: $\{\bar{W}_I, \bar{W}_C, \bar{W}_A\}$ of the representation networks $\{I(\cdot), C(\cdot), A(\cdot)\}$ for continuous case.

Further, we implemented 50 replicated experiments to evaluate the qini score for the binary case and the $\epsilon_{\text{PEHE}}$ for the continuous case. The results are summarized in Table 1.

**Ablation Study**    To validate the contribution of the adversarial modules to constrain $A(\boldsymbol{X}) \perp T$ and $I(\boldsymbol{X}) \perp Y|\{C(\boldsymbol{X}), T\}$, we implement extra experiments that removes the corresponding components.

(a) Figure 6a shows the resulted $\{\bar{W}_I, \bar{W}_C, \bar{W}_A\}$ after removing $h_{A \to T}(\cdot)$ and the related loss components. Compared to 5a, the model could not properly distinguish $A$ from $C$, where both $\bar{W}_C$ and $\bar{W}_A$ had nonzero weights on $X_{17} \sim X_{48}$.

(b) Figure 6b shows the results after removing $\{h_{C \cup T \to Y}(\cdot), h_{C \cup I \cup T \to Y}(\cdot)\}$ and the related loss components. Compared to 5a, the model performed worse in distinguishing $I$ from $C$, where $\bar{W}_I$ had more nonzero weights on $X_{17} \sim X_{32}$.

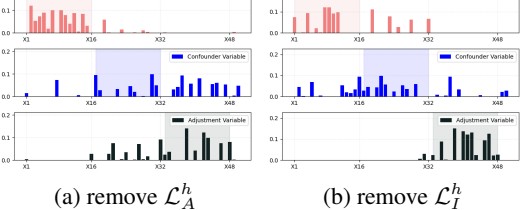

(a) remove $\mathcal{L}_A^h$        (b) remove $\mathcal{L}_I^h$

Figure 6: $\{\bar{W}_I, \bar{W}_C, \bar{W}_A\}$ for the ablation experiments.

## 5.3 EXPERIMENTS ON REAL DATASETS

IHDP (Infant Health and Development Program) is a widely adopted semi-synthetic benchmark dataset in the causal inference literature. Hill (2011) removed a non-random subset of the treated units in the original RCT data to generate an observational dataset with confounded treatment. IHDP dataset contains 747 instances (608 control and 139 treated) with 25 covariates. Outcomes were continuous and generated by the NPCI (Non-Parametric Causal Inference) package (Dorie, 2016). We use the dataset provided and used by Johansson et al. (2016) that includes 100 realizations. Coupon dataset is a large-scale production dataset from a coupon distribution scenario in JD, a leading E-commerce platform in China. In this scenario, 7 different values of the coupon (ranging from 1 to 5.5) were assigned to customers at the cashier interface to attract the customers to use a certain channel of payment. The treatment is continuous (coupon value) and the outcome is binary (whether the customer chose the payment channel that the coupon works for). In this case, the training set is from observational data and the evaluation dataset is from RCT. To evaluate the ranking performance of the estimated ITE, we calculate the qini score for each different coupon values paired with the control respectively and then take the average. All the numeric results are summarized in Table 1, where the values in parentheses are the standard errors calculated from the replicated experiments.

| Model | Synthetic Dataset | | Real Dataset | |
|---|---|---|---|---|
| | Binary Case qini score | Continuous Case $\epsilon_{\text{PEHE}}$ | IHDP data $\epsilon_{\text{PEHE}}$ | Coupon data qini score |
| CFR-MMD | 0.225 (0.018) | 0.0373 (0.0026) | 0.795 (0.078) | 0.0379 (0.0027) |
| CFR-WASS | 0.227 (0.015) | 0.0371 (0.0023) | 0.798 (0.058) | 0.0335 (0.0029) |
| CFR-ISW | 0.231 (0.019) | 0.0356 (0.0035) | 0.715 (0.102) | 0.0356 (0.0035) |
| DR-CFR | 0.268 (0.023) | 0.0363 (0.0032) | 0.789 (0.091) | 0.0401 (0.0011) |
| TEDVAE | 0.279 (0.020) | 0.0339 (0.0036) | 0.587 (0.089) | 0.0403 (0.0021) |
| DeR-CFR | 0.315 (0.018) | 0.0354 (0.0030) | 0.529 (0.068) | 0.0412 (0.0016) |
| ADR | **0.347** (0.020) | **0.0329** (0.0024) | **0.503** (0.072) | **0.0465** (0.0013) |

Table 1: Model performance evaluated by $\epsilon_{\text{PEHE}}$ on the synthetic dataset with continuous $y$ and the IHDP dataset, and evaluated by the qini score on the synthetic dataset with binary $y$ and the Coupon dataset. For $\epsilon_{\text{PEHE}}$, smaller value is better. For qini score, larger value is better.

## 6 CONCLUSION AND DISCUSSION

In this paper, we propose the ADR algorithm to learn decomposed representations for the ITE estimation, which has a wide application scenario including both categorical and numerical treatment. The empirical results show that the ADR algorithm is able to learn the correct decomposition and shows an advantageous performance in the ITE estimation compared to the state-of-the-art methods. The proposed ADR algorithm is guided by a preliminary theoretical analysis, where we show that the variables decomposition can be sufficiently confined by a series of probabilistic conditions and can be learned by an adversarial manner. Meanwhile, we believe the theoretical analysis is helpful to motivate other practical algorithms along this way (e.g. the algorithm that does not require such an adversarial training process and hence it is easier to get parameters to converge).

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
