# OpenReview forum: "Adversarial Learning of Decomposed Representations for Treatment Effect Estimation"
_ICLR.cc/2024/Conference — Submitted to ICLR 2024_

### Official Review · Reviewer_912z · 2023-10-18

**Soundness:** 2 fair
**Presentation:** 2 fair
**Contribution:** 2 fair
**Rating:** 3
**Confidence:** 5

**Summary:**

The main contributions of this paper are the ADR algorithm for decomposed representations in ITE estimation, a precise definition of variables decomposition, and theoretical analysis showing the benefits of this decomposition approach, including the variance lower bound of the CATE estimand. The ADR algorithm demonstrates its effectiveness through empirical validation and can be applied to a variety of treatment data types.

**Strengths:**

(1)The paper introduces the concept of $\mathbf{I, C, A}$ based on causal graphs and proves that this decomposition can be identified from observational data.

(2) A novel ADR algorithm is proposed, leveraging adversarial modules to ensure independence and conditional independence relations.

(3) This ADR algorithm is applicable to both categorical and numerical treatments and is supported by both theory and empirical results.

**Weaknesses:**

**Presentation**: There are many unclear statements. For example, `The ITE refers to $Y_i(t) − Y_i(0)$.' why do not write it as $Y_i(1) − Y_i(0)$? Eq.(1) is only presented for a binary treatment. How to define ITE or CATE for other types of treatments?


**Novelty**: The use of decomposed representation for identifying adjustment sets in causal inference has been previously explored in the literature. This paper likely builds upon existing methods and concepts while potentially introducing novel insights or improvements. In essence, several conclusions in the article may have already been substantiated. Additionally, the manuscript does not reference the literature that employs sufficient dimension reduction for learning the adjustment set.

**Contribution**: The conclusion in Theorem 3.2 has been proved by previous work[1,2], and both works also allow latent variables. So, the developed ADR can be regarded as a restricted version of the implementation of these two works. Therefore, the contributions of the work is not high enough for ICLR.

[1] Entner D, Hoyer P, Spirtes P. Data-driven covariate selection for nonparametric estimation of causal effects[C]//Artificial intelligence and statistics. PMLR, 2013: 256-264.

[2] Cheng D, Li J, Liu L, et al. Local search for efficient causal effect estimation[J]. IEEE Transactions on Knowledge & Data Engineering, 2022 (01): 1-14.

**Questions:**

Q1, `To deal with the issue, the common practice is to introduce pre-treatment covariates such that {Y (t)|x} =d {Y |t, x} (ignorability assumption).' Is it correct? If there are only pre-treatment covariates, it implies that there are no descendants of both $T$ and $Y$ in the set of covariates. How can we ensure $ignorability$ hold?

Q2, Eq.(2): `E[Y(t)|x]=E[Y(t)|x,T =t]=E[Y|x,T =t]', Can we really transform the potential outcome prediction problem into a supervised learning problem?


Q3. For the causal DAG in Fig. 1 (b),  does ADR also apply when $X_1$ is an unobserved variable.

---

> ### Author Response · Authors · 2023-11-20
>
> - Q1 [ignorability assumption]
>
> i) To our understanding, we believe that *"introducing all the pre-treatment covariates"* is a safe choice, but may not be an efficient one.
> Actually, the quoted sentence was referenced from [1]. We have also mentioned this in the paper (the 1st line of page 2).
> The full discussion can be accessed from https://ftp.cs.ucla.edu/pub/stat_ser/r345-sim.pdf.
>
> ii) *If there are only pre-treatment covariates, it implies that there are no descendants of both $T$ and $Y$ in the set of covariates. How can we ensure ignorability hold?*\
> The **ignorabity** assumption means that $\boldsymbol{X}$ blocked all the \textit{\bf{back-door paths}} from $T$ to $Y$ in the form $T\leftarrow \cdots Y$.
> Therefore, the descendants of $T$ and $Y$ are not necessary to block such paths.
>
>
> References:\
> [1] DB Rubin. Author’s reply (to judea pearl’s and arvid sjölander’s letters to the editor). Statistics in
> Medicine, 28:1420–1423, 2009.
>
>
>
> - Q2 [identification of E[Y(t)|x]]
>
> Sure, under the ignorability assumption, $\mathbb{E}[Y(t)|x]=\mathbb{E}[Y|x, t]$, which is the common practice in literature of treatment effect estimation.
> However, we care about the potential outcome for all the units, instead of the ones that receive treatment $T_i=t$.
> That is, we need to estimate the counterfactual outcome $Y_i(t)$ for the units with $T_i\neq t$
> This is why the inference of potential outcomes (and also the CATE) is more challenging than the common supervised learning problem.
>
>
> - Q3 [for the case with unobserved variables]
>
> Thanks for the question.
> In this case, since $X_1$ is not a confounder, ADR can be applied when $X_1$ is unobserved.

---

> > ### Comment · Reviewer_912z · 2023-11-21
> >
> > Thank you for your rebuttal.
> >
> >
> > (1) Regarding the assumption of ignorability, it is advisable to adhere to the standard assumption. The assumption of pretreatment alone is insufficient for establishing ignorability, as is well understood in the field.
> >
> > (2) Regarding  `E[Y(t)|x]=E[Y|x,T =t]',  the notation is confusing. E[Y|x, T =t] is a standard supervising learning model in your current notation. So it would be better to make this clear.
> >
> > (3) In fact, there is no confounding bias between $T$ and $Y$ in Fig. 1 (b) since the back-door path $T<-X_1->X_2<-X_3->Y$ is naturally blocked by the empty set. How can ADR remove the information of $X_2$ from the adjustment set $\mathbf{C}$ if $X_1$ is unobserved or $\{X_1, X_3\}$ are unobserved (M-bias)?
> >
> > Note that in Fig. 1(b), $X_2$ is not an IV based on the standard definition of IV. Hence, it would be better to change another name for this.

---

> > > ### Author Response · Authors · 2023-11-23
> > >
> > > Thank you for the response
> > >
> > > (1) Perhaps we need to mention that the *Ignorability Assumption* ($Y(t)\perp T|\boldsymbol{X}$) is the standard assumption adopted in the literature, which is also refered to as the "Unconfounded Assumption". Please kindly refer to the Def 3.6 (Unconfounded Assignment) on page 38 in [1].
> > >
> > >
> > > (2) This corollary is also a standard derivation in the literature of causal inference (under the Potential Outcomes Framework). Please kindly refer to Chap 12.4.2  -- *Model-Based Imputation* in [1].
> > >
> > > [1] Guido W. Imbens and Donald B Rubin. Causal inference for statistics, social, and biomedical
> > > sciences: An introduction. Cambridge University Press, 2015

---

### Official Review · Reviewer_sLak · 2023-10-23

**Soundness:** 3 good
**Presentation:** 2 fair
**Contribution:** 3 good
**Rating:** 5
**Confidence:** 4

**Summary:**

**Post-rebuttal update**: I maintain my score but add my final reply to the author(s).

*On Thm 3.1* It seems the quotation agrees with my understanding, and I guess the "their" does not refer to $\hat\theta_n$. All in all, do you agree that this result is not essential for the method? If so, I still suggest moving it to the Appendix. Otherwise, you should explain its importance together with clearer writing.

*On Def 3.1* I believe we agree that there are admissible sets diff from yours and might provide better finite sample performance. So you need to admit and explain this, particularly because your defs of I, C, A are diff from usual.

I still do not agree this method is theoretically guaranteed in a strict sense. You may want to explain in what sense you mean it.

**End update**

As for the problem of treatment effect estimation under unconfoundedness, the paper proposes to decompose the observed covariates into three disjoint sets, roughly corresponding to the usual concepts of instrumental, confounding, and adjustment variables. The variables are defined in graphical terms, and then reduced to independence relationships, by Theorem 3.2. An adversarial learning approach is proposed to induce the independence. Experiments show the benefits of the method.

**Strengths:**

The separation of covariates into three disjoint sets using Def 3.1 is an interesting idea.

The main theoretical results seem correct (proofs not checked, but I have my own (rough) proofs and cannot come up with counterexamples).

Experiments show the representations are practically decomposed, and the ablation study shows the usefulness of the theoretical ideas.

**Weaknesses:**

**Some problematic theoretical developments and discussions**

*Th 3.1* (variance lower bound). The statement seems incorrect or has typo(s). For CATE, the bound should depend on the value of x, but your eq of V takes expectation on X. Moreover, for consistent estimators, V should depend on n, and V → 0 as n → inf, but your V is a constant wrt n. Anyway, I don’t see this result has a strong relationship to the method (or else I will give a lower score), you could remove this result if you cannot fix it.

*Def 3.1* deviates from standard notions in the literature and also has practical limitations. For example,

- in your Fig 1a, if there is an X4 that is a parent of both X1 and X2, this is usually understood as a confounder and can improve the precision of estimation, but is excluded from your approach.
- your Fig 1b is actually the “M-bias” case (see model 7 [here](https://ftp.cs.ucla.edu/pub/stat_ser/r493.pdf)). X1 and X3 both satisfy the backdoor criteria and could be understood as confounders. Here, besides X3, both X1 and X2, which are excluded from your approach, could possibly improve the precision (though X2 alone is a bad control).

I suggest being clear that the definition is nonstandard, discussing and comparing it to usual notions (possibly in the Appendix). In particular, you should mention there are possible variables in your IVs that are good to add as controls.

See Questions for more comments.

**The method is theoretically motivated but *not* theoretically guaranteed.**

Prop 3.2 seems correct but the learning approach is not sufficient. Taking (i), I agree that independence means larger L_A than dependence, but, there can be many different functions A that give the independence. Worse, some A could take a confounder but “cleverly” through away the dependence on T. Similar comments apply to (ii). Could your theory rule out these concerns?

The ADR algorithm does not precisely enforce the required independence or even the approach in Prop 3.2, because L_A, L_C, L_I contain both prediction and adversarial terms, so the ADR is a trade-off but not a direct implementation of the theory. Moreover, training the losses together with hyper-parameters adds yet another layer of trade-off.

I suggest weakening the claims on this contribution.

**Questions:**

I will read the rebuttal and revised paper and raise my score to 6 if the issues/questions in Weaknesses are addressed. Some further points are as below.

Prop 3.1 (i) I think we can say “either…or…” which is stronger than simply “or.” Also, it is safer to say “X \indep T and X \indep Y” which is weaker than the joint independence and seems enough.

It is confusing to only stress C in the last statement of Th 3.2. In fact, A may also be sufficient, as in your Fig 1b.

The comments below Th 3.2 are confusing. It is an identification because the 3 sets of variables are determined by the observable joint distribution, through the conditional independence requirements. In fact, the definition of I/C/A implicitly assumes graphical structures, and you reduce the graphical structure to independence by *causal Markov and faithfulness assumptions*. Indeed, these *are* the “further assumptions” you also use.

Add experiments that directly evaluate identification and decomposition. Actually, Fig 3 and 5 show the method does not fully identify and decompose the covariates. Thus, it is meaningful to examine this more closely. For example, we could build several datasets with only one I, C, A respectively, and plot the learned I, C, A against the truth.

As to identifiable representation, the recent advance in using deep identifiable model (e.g., [1]) to estimate treatment effect (e.g., [2, 3]) is worth discussing in the related work.

[1] Khemakhem, Ilyes, et al. "Variational autoencoders and nonlinear ICA: A unifying framework." International Conference on Artificial Intelligence and Statistics. PMLR, 2020.

[2] Wu, Pengzhou Abel, and Kenji Fukumizu. "beta-Intact-VAE: Identifying and Estimating Causal Effects under Limited Overlap." International Conference on Learning Representations (2022).

[3] Ma, Wenao, et al. "Treatment Outcome Prediction for Intracerebral Hemorrhage via Generative Prognostic Model with Imaging and Tabular Data." International Conference on Medical Image Computing and Computer-Assisted Intervention., 2023.

Minor (did not affect the score):

It is bad to use the abbreviation ITE for the Individual-level Treatment Effect. Maybe you could use “ILTE” instead. Actually, “ITE” in your paper refers to both ILTE/CATE and eq1, which is the correct definition of ITE.

"Adjustment variables" usually mean the set of variables conditional on which the confounding is removed. Only in some ML papers do adjustment variables refer to those variables that affect Y but not T. This is another often-seen misnomer in the ML community.

The \mathcal(L) in Prop 3.2 should be a typo.

---

> ### Author Response · Authors · 2023-11-20
>
> Thanks for your efforts and valuable comments on our paper. We address your concerns below:
>
> - Q1 [questions on the Thm 3.1]
>
> Thanks for the interest in Thm 3.1. It seems like there might be some understanding due to this result.
>
> i) *Why the variance bound is constant w.r.t. $n$.*
>
> Please note that Thm 3.1  is about the \textit{asymptotic variance bound}, instead of the \textit{variance} of the estimation.
> 	The asymptotic variance refers to the variance of the limit distribution, instead of the limit of the variance.
> 	Please allow us explain this with a simple example.
> 	Suppose that $\{X_i\stackrel{i.i.d.}{\sim}N(\mu, \sigma^2)\}$ and $\hat{\mu}=\frac{\sum_i X_i}{n}$ is the estimation of $\mu$, then $\sqrt{n}(\hat{\mu}-{\mu})\stackrel{d}{\rightarrow} N(0, \sigma^2)$.
> 	In this case, the asymptotic variance of $\hat{\mu}$ is $\sigma^2$, and the variance of $\hat{\mu}$ is $\sigma^2/n$. Please kindly refer to [1] for the formal definition.
>
> ii) *Why the variance bound is not relevant to $x$*.
>
> Suppose that $\tau_0$ is the underling value, and $\widehat{\tau}(x)$ is the estimation. Then the variance of $\tau(x)$ refers to $\mathbb{E}[\tau_0-\widehat{\tau}(X)]^2$, a real number. The conditional variance  $\mathbb{E}[\tau_x-\widehat{\tau}(X)]^2$ is a function of $X$.
>
> References:\
> [1] Van der Vaart A W. Asymptotic statistics[M]. Cambridge university press, 2000.
>
>
> - Q2 [questions on Def 3.1]
>
> The terms "Instrument variables" and "Adjustment Variables" were borrowed from the notations from DeR-CFR [2].
> We admit that such names may run at a risk of deviating from the standard notions.
>
> i)  *``in your Fig 1a, if there is an $X_4$ that is a parent of both $X_1$ and $X_2$, this is usually understood as a confounder and can improve the precision of estimation, but is excluded from your approach."*
>
> In this case, $P(Y|X_2, X_3, do(t))=P(Y|X_2, X_3, X_4, do(t))$, which means that adding $X_4$ is spurious after including $\{X_2, X_3\}$.
> More generally, we may prove that $p(Y|do(t), \boldsymbol{X})=p(y|do(t), \boldsymbol{C}, \boldsymbol{A})$. We have added this result in Prop 3.2.
>
> ii) *``your Fig 1b is actually the “M-bias” case (see model 7 here). X1 and X3 both satisfy the backdoor criteria and could be understood as confounders. Here, besides X3, both X1 and X2, which are excluded from your approach, could possibly improve the precision (though X2 alone is a bad control)."*
>
> In the example of Fig 1b, there is no unblocked back-door path(s) from $T$ to $Y$. Therefore, we could not agree that $X_1$ and $X_3$ could be called as confounders. However, both $X_1$ and $X_3$ are admissible set in [3] that satisfies $p(y|x_1, do(t))=p(y|x_1, t)$ and  $p(y|x_1, do(t))=p(y|x_1, t)$.
>
>
> References:\
> [2] Anpeng Wu, Junkun Yuan, Kun Kuang, Bo Li, Runze Wu, Qiang Zhu, Yueting Zhuang, and Fei
> Wu. Learning decomposed representations for treatment effect estimation. IEEE Transactions on Knowledge and Data Engineering, 35(5):4989–5001, 2022.\
> [3] Pearl J, Paz A. Confounding equivalence in causal inference[J]. Journal of Causal Inference, 2014, 2(1): 75-93.
>
>
> - Q3 [questions on Prop 3.2]
>
> The learning approach is a direct implementation of Prop 3.2.
> The $\hat{h}\_{A\rightarrow T}(\cdot)$ in Prop 3.2 corresponds to the ancillary predictor $h_{A\rightarrow T}(\cdot)$.
> Prop 3.2 claims when $\mathcal{L}_A$ is maximized when $A\perp T$, so we maximize $\mathcal{L}_A$ in the learning process.
> Could you please be more specific on the question *``Worse, some A could take a confounder but “cleverly” through away the dependence on T. Similar comments apply to (ii). "*.

---

> ### Comment · Reviewer_sLak · 2023-11-20
>
> Thanks for the rebuttal. However, most of my concerns remain.
>
> **On Thm 3.1**
>
> i) In your example, I would call $\sigma^2/n$ as the “asymptotic variance” of $\hat\mu$, while $\sigma^2$ is the asymptotic variance of $\sqrt{n}\hat\mu$. Could you give the relevant page number(s) to the book you referred?
>
> ii) Here, the estimator is $\hat\tau$, a function of a *fixed* (non-random) x, while an estimator is by definition a function of a *random* sample. Thus when we talk about the variance of the estimate $\hat\tau$, we see $\hat\tau$ as an RV depends on the sample, and the expectation should be taken on the sample (but not the RV X). Also, I do not understand your distinction between $\tau_0$ and $\tau_x$, actually, when you say “the variance of $\tau(x)$”, the conditional on x is implicitly introduced. You might refer to another thing, but this confusion should be addressed.
>
> **On Def 3.1**
>
> i) That X4 can be removed means that X2 and X3 are sufficient for controlling the confounding, but *not* that X4 is spurious. Including X4 can improve the precision of estimation, similar to those settings where variables are “possibly good for precision” in [1].
>
> ii) Please refer to [1], which says the path T-X1-X2-X3-Y is a back-door path (conditional on X2). Specifically, the sets {}, {X1}, {X3}, {X1, X2},  {X2, X3}, {X1, X3}, {X1, X2, X3} all satisfy the back-door criterion, that is, all the subsets of {X1, X2, X3} *except* {X2}.
>
> [1] Cinelli, Carlos, Andrew Forney, and Judea Pearl. "A crash course in good and bad controls." *Sociological Methods & Research* (2022): 00491241221099552. **Do read this one** (linked in my original review) and you will come back and thank me. (Pearl himself is an author and referred to it several times on Twitter.)
>
> **On Prop 3.2**
>
> I do not agree “The learning approach is a direct implementation of Prop 3.2”. There was a typo in my review and I should have written “*throw away the dependence on T*”, by which I meant, that for a confounder C, there can be an A such that A(C) \indep T.

---

> > ### Author Response · Authors · 2023-11-23
> >
> > Thanks very much for your response.
> >
> > **On Thm 3.1**
> >
> > i) The term *asymptotic variance* first appear on page 3 (the 2nd paragraph). It reads "*Third, their asymptotic variance, the variance of the limit distribution of $\sqrt{n}(\hat{\theta}_n - \theta)$ is minimal*".
> >
> > ii) Sorry for the inconsistency of the notation. I should write $\tau_x$, the ground truth CATE. The variance is not conditional on $X$, I will address this confusion in the paper.
> >
> > **On Def 3.1**
> >
> > Thanks for recommending the literature and We have read it through.
> >
> > i) I could not agree that $X_4$ is similar to the setting of Model 8 and 13 (*possibly good for precision*). Because in our case, $p(y|x_2, x_3, do(t)) = p(y|x_2, x_3, x_4, do(t))$, while in Model 8, $p(y|do(x), z)\neq p(y|do(x))$.
> > Although explanation by graph is intuitive and inspiring, I suppose justifying by the interventional distribution is also a plausible and clear way.
> > Besides, as [1] mentioned in the ``Model 8", "*controlling for $Z$ reduces the variation of the outcomes $Y$ and helps to improve the precision of the ACE estimate in finite sample*". I find this this motivating because in this case controlling for $Z$ is not necessary (in terms of bias), but benefits the precision in finite sample.
> >
> > ii) I agree that all the subsets except from $\{X_2\}$ are admissible sets to block the back-door paths. Meanwhile, I also think it sufficies to include $X_3$ in modeling the ITE.
> >
> >
> > **On Prop 3.2**
> >
> > Thanks for the futher explanation. If $A(C)\perp T$, it means that the information of $C$ may not be catched by $A(\cdot)$. Perhaps this can be appreciated by the results in Figure 3, we investigate the $\overline{W}_I$, $\overline{W}_C$, $\overline{W}_A$ to justify the owenership of the *learned* decomposed representations.
> > By saying that "a direct implementation of Prop 3.2", we intend to express that the loss function and the iterative learning process is designed according to Prop 3.2.
> > For the ADR algorithm, Prop 3.2 plays the same role as Prop 1 (global optimality) for GAN.

---

### Official Review · Reviewer_8Xzt · 2023-11-01

**Soundness:** 3 good
**Presentation:** 2 fair
**Contribution:** 3 good
**Rating:** 5
**Confidence:** 3

**Summary:**

This paper discusses efficient estimation of Conditional Average Treatment Effects (CATE), working primarily in the case where the covariate set is high-dimensional and contains different kinds of pre-treatment covariates (e.g., confounders, IVs, variables only affecting the outcome).

**Strengths:**

See below for a contextual discussion of strengths and perceived weaknesses.

**Weaknesses:**

In my view, this paper strikes me as overall well-written and motivated (albeit somewhat heavy on notation which could limit its broader impact). The assumptions used in the paper are standard for observational inference (which I view as a strength of the paper). The point that variance bounds are affected by pre-treatment covariate number and that distinguishing between kinds of pre-treatment estimation variance bounds in an effort to improve the bound is intriguing, as is the notion that we can distinguish between pre-treatment covariates of different types in an identified manner.

My main comments concern the ability of readers to evaluate the contribution of the paper in view of the literature. For example, what is the relationship between the work on semiparametric efficiency bounds in effect estimation with some of discussion here. The literature on, e.g., semi-parametric efficiency is often focused on ATE (as opposed to CATE estimation as here), but even a discussion of the efficiency of the approach here for the ATE vs. in that setting would be most informative for this reviewer.

On a related note, the paper could further improve its contribution by evaluating observational ATE recovery against some of the most commonly used methods for that (e.g., doubly robust methods and something simple like inverse propensity score weighting). If readers can see that the proposals here by improving observational CATEs also improve observational ATEs (which have extremely broad applicability in existing applied work, much more than observational CATEs), the paper's contribution would be enhanced.

On another note, the decomposition of I(X), C(X), and A(X) would be extremely useful in practice. However, one limitation is that in any given experiment, we cannot know/validate for sure (and if there is good a priori reason to suspect a covariate is an I, C, or A adjustment could proceed directly with that knowledge). Nevertheless, if the authors could obtain a case (perhaps from, e.g., the biological context where biophysical relations are approximately known) where the decomposition provides useful information to the investigator, I would think the contribution would also be improved. By the way, it would be very convincing if the approach here was somehow better than using a priori knowledge of the decomposition directly.

A few small comments:

(1) Not to sound pedantic, but the writing at the sentence/paragraph level is somewhat stronger than across sections. For example, there is much discussion of the variance bound in the theory section, but this emphasis disappears later on. The paper can sometimes feel disjointed (as if separate contributions are fused).

(2) I would edit the "Algorithm 1" text to remove the reference to (I believe) the specific optimizer Adam. Optimizers will come and go with time and presumably, the contribution here is more general, and other optimizers would work as well in principle.

**Questions:**

One question concerns whether the authors intend investigators to actually examine the inferred decomposition of X, or whether the motivation is mainly or exclusively efficient CATE estimation.

**Details Of Ethics Concerns:**

No ethics concerns.

---

> ### Author Response · Authors · 2023-11-20
>
> Thanks a lot for your efforts and valuable comments on our paper. Our answers are as below:
>
> - Q1 [the purpose of decomposition]
>
> Thanks for the question.
> We admit that the original purpose is to facilitate a more efficient CATE estimation.
> However, we believe that such decomposition is beneficial for further analyses on the covariates, especially in the case where interpretation is important.
>
> - Other Comments
>
> As suggested, we haved revised Algorithm 1 remove the reference to  the specific optimizer Adamin the updated version. Thanks for the kind suggestion.

---

### Official Review · Reviewer_Yi3t · 2023-11-01

**Soundness:** 3 good
**Presentation:** 2 fair
**Contribution:** 3 good
**Rating:** 5
**Confidence:** 4

**Summary:**

This paper focuses on the disentanglement of instrumental variables (I), adjustment variables (A), and confounders (C) (distinguished according to their dependence on the treatment and outcome variables) from covariates for causal effect inference.
They provide an identifiable definition of these variables and a method based on adversarial training in which two discriminators predict the treatment and outcome variables from A and I, respectively, and the representation extractors for A and I counter them.

**Strengths:**

* Since the representation balancing for CATE estimation is pointed out as not capturing the whole CATE estimation errors in literature, representation decomposition is a promising direction as a response.
* In this context, the first identifiable formulation of representation decomposition through an adversarial formulation would be a very mainline approach.
* A simulation-based experiment clearly illustrates its superiority in disentanglement performance compared to some existing methods.

**Weaknesses:**

1. The aim is not clear. The disentanglement itself seems to be the aim, and it is not clear how it contributes to the accuracy of the CATE (see Question 1).
1. The design of the loss function is somewhat heuristic and a logical explanation or guarantee is insufficient (see Question 2).
1. The adversarial joint objective is not in a convex-concave formulation, which means there is no guarantee of convergence. Intuitively, it seems very unstable.
    * Are there any existing studies of such a formulation that *maximizes* the loss function such as the MSE?
    * While maximizing the MSE by the adversary is easily accomplished by making the predictions infinity, it seems to be difficult to predict it accurately.
    * It may be helpful to show realistic convergence using a learning curve.

**Questions:**

1. What is the purpose of the decomposition? The original purpose was to combine weighting only w.r.t. the confounders in DR-CFR, in my understanding. Confounder variables should be limited to necessary ones to alleviate the estimation variance due to extreme weights. The proposed method does not use weighting and thus I am confused about its aim.
    1. A possible reason for the above question is to limit the input, i.e., excluding instrumental variables from the input of the predictor, as suggested in Thm 3.1. Although, Thm 3.1 is only about the variance lower bound and I am not sure if that is dominant or critical in the estimation error. Does excluding I(x) from the input of the predictor really have a decisive impact? Any theory about the whole risk bound of the proposed method, or an ablation experiment on the "with-I(x) model" $f_{C\cup A\cup I\cup T \to Y}$ instead of $f_{C\cup A\cup T \to Y}$ might provide empirical evidence.
1. Why $L_A$ does not include the accuracy of $f_{C\cup A\cup I \to Y}$? A(x) is input to $f_{C\cup A\cup I \to Y}$, but the gradient for the connection is stopped. Does not this have any negative impact on the whole design of the optimization procedure?

Minors:

* P4 Theorem 3.2 stats -> states
* P4 Hassanpour & Greiner (2020). -> [Hassanpour & Greiner (2020)].

=== EDIT after reading the reponse===

I would like to keep rating because I think the current structure is insufficient in the following points, although I recognize a certain value in that they newly define the I/C/A definitions and derive a principled method.
My concerns are summarized two-fold (and not solved yet).

1. Motivation: the existing decomposition-based methods are weighting-based methods, whereas the proposed method is not, and does not perform full bias removal at the loss function level (thus, e.g., not guaranteed to be consistent when misspecified). Therefore, it does not seem appropriate to position this as this research stream.
- Rather, it should be positioned in the stream of simple modeling methods, such as T-Learner, which is not organized and written as such, and thus lacks a motivation argument to claim value in that context.
- Also, when viewed simply as a SOTA CATE estimation method, it lacks sufficient statistically significant results.

2. Soundness: the method is considered as a heuristic alternate update and not formalized as a joint objective; thus the optimization can be unstable (maybe it does not converge).
- Unlike GANs, it is not formalized as a zero-sum game that provides some convergence guarantee.

---

> ### Author Response · Authors · 2023-11-20
>
> Thanks a lot for your efforts and valuable comments on our paper. We address your concerns below:
>
> - [Q1. purpose of the decomposition]
>
> We agree that it is important to answer the question for implementing such decomposition for the ITE estimation.
> It is absolutely correct that the purpose is to `` limit the confounder variables to necessary ones to alleviate the estimation variance due to extreme weights".
> However, it should be noted that, even we do not adopt the inverse probability weighting method, the estimation variance also increase with the aggravation of data imbalance as Thm 3.1 suggests.
> We may have a further understanding by reviewing the development of the literature.
>
>
> Firstly, the balanced representation leaning algorithms [1, 2] were proposed to reduce the data imbalance caused by selection bias.
> Except from the empirical evidence, [2] explained the benefit from the view of generalization bound.
>
> Secondly, [3] pointed out that leaning the balanced representation alone may overlook the necessary confounding information and hence should learn the decomposed representation. However, the loss terms of DR algorthm in [3] was not complete and hence was not able to learn the decomposed representations. [4] proposed the DeR algorithm to realize the decomposition learning under the situation of binary treatment and response.
> To deal with the case for continuous treatment and adapt to both categorical and continuous response, we analyzed the theoretical property of the decomposed representations and propose the ADR algorithm.
>
>
>
>
> References:\
> [1] Fredrik Johansson, Uri Shalit, and David Sontag. Learning representations for counterfactual
> inference. In International Conference on Machine Learning, pp. 3020–3029. PMLR, 2016.\
> [2] Uri Shalit, Fredrik D Johansson, and David Sontag. Estimating individual treatment effect: generalization bounds and algorithms. In International Conference on Machine Learning, pp. 3076–3085.
> PMLR, 2017.\
> [3] Negar Hassanpour and Russell Greiner. Learning disentangled representations for counterfactual
> regression. In International Conference on Learning Representations, 2020\
> [4] Anpeng Wu, Junkun Yuan, Kun Kuang, Bo Li, Runze Wu, Qiang Zhu, Yueting Zhuang, and Fei
> Wu. Learning decomposed representations for treatment effect estimation. IEEE Transactions on
> Knowledge and Data Engineering, 35(5):4989–5001, 2022.
>
> - [Q2. the design of $\mathcal{L}_A$]
>
> We have added the loss of $f\_{A\cup T\rightarrow Y}(\cdot)$ in $\mathcal{L}\_A$ to learn the representation that are predictive to $Y$.
> The loss were based on the result in Thm 3.2.
> We minimize the loss of $f\_{A\cup T\rightarrow Y}(\cdot)$ to realize $\boldsymbol{A}\not\perp Y$ and
> maximize the loss of $h\_{A\rightarrow T}(\cdot)$ to realize $\boldsymbol{A}\perp T$.
>
>
> - [Q3. the convergence guarantee]
> Thanks for the insightful question.
>
> i) ``*Are there any existing studies of such a formulation that maximizes the loss function such as the MSE?*"
>
> A similar formulation is that in GAN [5], the generator is updated by maximizing the cross-entropy loss (the discriminator us undated by minimizing the cross-entropy loss). In this case, the optimal predictor has an explicit solution and such that the convergence can be readily proved (see Sec 4.2 in [5])
>
> ii) ``*While maximizing the MSE by the adversary is easily accomplished by making the predictions infinity, it seems to be difficult to predict it accurately.*"
>
> Please kindly note that in maximizing the loss $\\mathcal{L}\_A^h:=\sum_i l(h\_{A\\rightarrow T}(A(\\boldsymbol{x}_i)), t_i)$,
> the $h\_{A\rightarrow T}(\cdot)$ is fixed and supposed to be (close to) the optimal predictor given $A(\cdot)$.
> 	As is suggested by the proof of Prop 3.2,
> 	$\mathcal{L}\_A^h\approx Var[\mathbb{E}[T|A(X)]]=Var(T)-\mathbb{E}[Var[T|A(X)]]$ that is less equal than $Var(T)$ and is maximized when $A(X)\perp T$.
> 	Practically, we restrict the values of representation networks by the $\mathcal{L}_O$ in eq (7) such that $(\sum \overline{W}[k]-1)^2$ is close to zero.
>
> iii) ``*It may be helpful to show realistic convergence using a learning curve.*"
>
> Thanks for the suggestion. We have checked the tensorboard results saved in the training process. For example, in the case of Synthetic Dataset (continuous treatment). It can be seen that $\mathcal{L}_A^h$ has an oscillation at the very beginning and then starts to increase and end at a steady level after 2.5k steps.
> 	We have added the relevant results in the Supplementary.
> 	Unfortunately, we have not yet come up with a way to theoretically prove the convergence for continuous treatment.
> 	We suppose that the result would be similar to Proposition 2 in GAN [5].
>
>
> References:\
> [5] Ian Goodfellow, Jean Pouget-Abadie, Mehdi Mirza, Bing Xu, David Warde-Farley, Sherjil Ozair,
> Aaron Courville, and Y. Bengio. Generative adversarial nets. In Neural Information Processing
> Systems, 2014.

---

### Official Review · Reviewer_zHRm · 2023-11-07

**Soundness:** 2 fair
**Presentation:** 3 good
**Contribution:** 1 poor
**Rating:** 3
**Confidence:** 3

**Summary:**

The paper introduced a new decomposed representation learning method for conditional average treatment effect (CATE) estimation. It is based on a theoretic property that all the covariates in the valid adjustment set can be either instrumental variables, adjustment variables, confounders, or background noise variables, and that this this decomposition is identifiable from the observational distribution. The paper then develops an adversarial learning technique to decompose the covariates into three categories of instrumental variables, adjustment variables, and confounders. The authors compare their method, namely, adversarial learning of decomposed representations (ADR), with the existing representation learning baselines for CATE estimation on several synthetic and semi-synthetic benchmarks.

**Strengths:**

The paper is clearly written and well-structured. I found the theoretic results of the paper regarding the decomposition of the insightful and important for representation learning for CATE. For example, I appreciate that the authors provided formal identification guarantees for the decomposed representation, i.e., Prop. 3.1 and Theorem 3.2. Also, the experimental results on decomposing, i.e., Figures 3-5, are very informative.

**Weaknesses:**

There are several issues in this paper:
  1. Error in derivations. I spotted two issues. First, Theorem 3.2 claims that $\mathbf{C}$ is a valid set $\mathbf{X}’$ in the definition of the instrumental variables. On the other hand, by looking at the example in Fig. 1 (b), $\mathbf{C} = \varnothing$, but $X_2 \notindependent Y \mid T$. Second, there seems to be an erroneous statement in the proof of Prop. 3.2, that the equality in the expectation $\mathbb{E} (T \mid A(X) ) = \mathbb{E}(T)$ implies the independence, $T \independent A(X)$, which is not true, if $T$ is continuous. Specifically, there could be inequalities wrt. to higher moments. Those two issues are further very important for the correct implementation of the ADR.
2. Novelty. The implementation of the decomposed representation learning with adversarial representations, namely, ADR, was already proposed in [1], and this work is not even mentioned in the related work or included as a baseline. Therefore, the paper has only a marginal contribution.
3. Implementation and tuning. Some details are missing on the implementation of the baselines, e.g., the dimensionalities of the representations.  Also, the authors did not provide any details on how to choose the dimensionalities of the decomposed representations in their method, which is a very important issue in practice, e.g., for the IHDP benchmark. Therefore, it is impossible to say, whether the empirical evaluation was fair.

I am open to raising my score if the authors address all my concerns.

References:
[1] Chauhan, V. K., Molaei, S., Tania, M. H., Thakur, A., Zhu, T., & Clifton, D. A. (2023, April). Adversarial de-confounding in individualised treatment effects estimation. In International Conference on Artificial Intelligence and Statistics (pp. 837-849). PMLR.

**Questions:**

See the section on weaknesses.

---

> ### Author Response · Authors · 2023-11-20
>
> Thanks for your efforts and valuable comments on our paper. We address your concerns below:
>
> - [Q1] Thanks a lot for pointing out the questionable part in our work.
>
> i) As for the claim that $\boldsymbol{I}\perp Y|T\cup \boldsymbol{C}$ in Thm 3.2, we admit the definition for $\boldsymbol{I}$ (instrumental variables) was inappropriate.
> We should rule out the "collider" variables in the definition of $\boldsymbol{I}$.
> Under this updated definition, $\boldsymbol{I}=\{X_1\}$ for Figure 1(b). We have revised the paper accordingly, and also corrected the corresponding proof for Thm 3.2 in the Supplementary.
>
> ii) We apologize that the careless statement in the proof of Prop 3.2 caused your confusion.
> We have changed \textit{"when $\mathbb{E}(T|A(X))=\mathbb{E}(T)$, i.e., $T\perp A(X)$"} as \textit{"when $T\perp A(X)$, we have $\mathbb{E}(T|A(X))=\mathbb{E}(T)$"}.
> It should be noted that Prop 3.2 claims that $\mathcal{L}_A$ is maximized when $A\perp T$, which is in align with the corrected statement.
>
> - [Q2] Thanks for recommending the interesting literature.
>
> We have to point out that, [1] is very different from our proposed ADR algorithm, although the title seems similar.
>
> Firstly, the SNnet+ proposed in [1]
> aims to maximize the loss of treatment classifier on top of the confounder representation $\boldsymbol{C}$.
> By contrast, we maximize the loss of treatment classifier for the adjustment representation $\boldsymbol{A}$.
>
> Secondly, the implementation is different.
> The SNnet+ added a gradient reversal layer to reverse the direction of the gradient $\frac{\partial L_t}{\partial W_c}$ in the back-forward pass.
> Our proposed ADR algorithm adopts an iterative training process that firstly trains the treatment classifier of $\boldsymbol{A}$ given the representation, and then fix the classifier and maximize the loss in updating the representation.
>
>
> Reference: \
> [1] Chauhan, V. K., Molaei, S., Tania, M. H., Thakur, A., Zhu, T., \& Clifton, D. A. (2023, April). Adversarial de-confounding in individualised treatment effects estimation. In International Conference on Artificial Intelligence and Statistics (pp. 837-849). PMLR.
>
> - [Q3] Implementation Details
>
> We adopted the same representation dimension (256) and the same structured MLPs for the predictors on top of the representations in the experiments.
> For a fair comparison, the values of such parameters were also in align with the compared literature [2]. The implementation details were added in the Supplementary (see \verb|Supplementary/code|).
> Thanks for pointing out this problem, we have
> added the relevant information in Section 5.
>
> References: \
> [2] Anpeng Wu, Junkun Yuan, Kun Kuang, Bo Li, Runze Wu, Qiang Zhu, Yueting Zhuang, and Fei
> Wu. Learning decomposed representations for treatment effect estimation. IEEE Transactions on
> Knowledge and Data Engineering, 35(5):4989–5001, 2022.

---

> > ### Comment · Reviewer_zHRm · 2023-11-23
> >
> > Dear authors, thank you for the explanations and the clarifications. Nevertheless, the following very important issues persist in the paper:
> >
> > - [Q1] (ii) So you agree, that in general, the equality of the conditional expectations does not imply independence? If yes, it seems to be wrong that ADR enforces the independence with the equality of the conditional expectations.
> >
> > - [Q2] I would expect SNnet+ to be included as a baseline in this case.
> >
> > Thus, I still tend to keep my score the same.

---

> > > ### Author Response · Authors · 2023-11-23
> > >
> > > Thank you for the response.
> > >
> > > **Q1**
> > >
> > > We agree that the equality of conditional expectation does not imply independence. That is,  the *equality of conditional expectation* is a necessary condition for *independence*.
> > >
> > > Therefore, we do think that ADR is appropriate to enforce the independence by constraining the equality of conditional expecations.
> > >
> > > Perhaps this can be explained by refering to the *Balanced Representation Learning* method CFR-net in [1] as an analogy. In [1], the balanced representation is learned by adding the IPM (Integral Probability Metreics) in the loss function.
> > >
> > > The reasoning is similar, when $\{\Phi |T=0\} \stackrel{d}{=}\{\Phi |T=1\}$, the $\text{IPM}(\Phi|T=0， \Phi|T=1)$ is zero. However, when the IPM is zero (*e.g., when we adop the MMD distance with linear kernel*), it does not suggest the exact equality in distribution.
> > >
> > >
> > > [1] Shalit U, Johansson F D, Sontag D. Estimating individual treatment effect: generalization bounds and algorithms[C]//International conference on machine learning. PMLR, 2017: 3076-3085.

---

> > > > ### Comment · Reviewer_zHRm · 2023-11-23
> > > >
> > > > I disagree. The equality of conditional expectations does not imply independence, especially when $A(X)$ is a continuous representation. Instead, we need to enforce the equality between conditional distributions, as this is the definition of conditional independence.

---

> > > > > ### Author Response · Authors · 2023-11-23
> > > > >
> > > > > Thanks for the response.
> > > > >
> > > > > We admit that ”*the equality of conditional expectations does not imply independence*“, as was explained in the former response and *claimed in the Prop 3.2*.
> > > > >
> > > > > Using the ”*the equality of conditional expectations*" can be regarded as a weaker but easier-to-implement substitute.

---

### Meta-Review · Area_Chair_ZNv7 · 2023-12-06

**Metareview:**

The paper proposes ADR algorithm to learn the decomposed representations and simultaneously estimate the treatment effect.

pros:
+ The paper studies an important problem of representation learning for causal estimation.

cons:
+ lack of sufficient theoretical foundations/justifications for the proposal
+ lack of sufficiently fair empirical comparisons with existing methods
+ concerns with derivations in theorems
+ lack of clear motivation of the decomposed approach

**Justification For Why Not Higher Score:**

lacking in both theoretical and empirical front

**Justification For Why Not Lower Score:**

NA

---

### Decision · Program_Chairs · 2024-01-16

Reject